# Fretting Fatigue Test and Simulation Analysis of Steam Generator Heat Transfer Tube

**Lichen Tang** [1,*] **, Hao Qian** [1] **, Chen Li** [1] **and Xinqiang Wu** [2]

[1]    Shanghai Nuclear Engineering Research and Design Institute Co. Ltd., Shanghai 200233, China
[2]    Institute of Metal Research, Chinese Academy of Sciences, Shenyang 110016, China
*    Correspondence: tanglichen@snerdi.com.cn

**Abstract:** In a steam generator, the heat transfer tubes are supported by the contact with support plates and anti-vibration bars. The two-phase flow flows over the tubes and causes a vibration when operating. In fatigue analysis, the heat transfer tube is simplified to a beam model, and the contacts with the support plates and the anti-vibration bars are simplified as simple-supported boundary conditions. This linear simplification improves the computational efficiency but cannot simulate the actual situation of the contact area. In consideration of this situation, in the actual analysis, a downwards modified S–N fatigue curve is used to envelop the fretting. For different materials and contact pressure, this modification needs to be obtained through experimental and computational analysis. In this paper, the effect of fretting on fatigue performance of heat transfer tube material 690 alloy is discussed by means of high cycle fretting fatigue test of tube specimen in room temperature air, low cycle fretting fatigue test of sheet specimen in high temperature water environment, and SWT (Smith–Watson–Topper) fretting fatigue predicting simulation, and the conservatism of design fatigue curve is discussed. It is shown that, in range of low cycle and high cycle, the fatigue strength is lower than the mean curve, but it can still be enveloped by the design curve of ASME (the American Society of Mechanical Engineers). However, under the condition of ultra-high cycle, the design curve of ASME can no longer envelop the effect of fretting on fatigue performance, so a further downward modification is necessary to ensure the safety of design.

**Keywords:** heat-transfer tube; fretting fatigue test; numerical simulation; 690 alloy; fatigue design curve

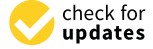

## 1. Introduction

Fretting is the motion of the micrometer magnitude amplitude of two contact surfaces, usually occurring on the contact surface of a vibratory environment [1]. Fretting can result in two main consequences: material loss on the surface and the reduction of fatigue strength of the structure, which is called fretting fatigue.

The heat transfer tube of the steam generator is the key component to isolate the boundary of the first and second loop, and its operation reliability is very high. It is not only the key component but also the weak part [2]. The rupture of the heat transfer tube will cause reactor coolant system small water loss accident to endanger the reactor safety, and the radioactive material leaked from the broken pipe will be released from the second loop to the environment. The stiffness of the thin-walled structure of the heat transfer tube is low, and the stiffness of the structure is improved by the support of the supporting plates and the anti-vibration bars. When the equipment is running, there is a slight collision and slip between the heat transfer tube and the supporting plate and the anti-vibration bar under the excitation of the flow induced vibration, which increases the possibility of the fretting damage and failure.

The heat transfer tube of steam generator is in a long-term vibration state under the fluid excitation, and the number of vibrations in its entire life will reach $10^{10}$ to $10^{12}$ cycles.

In the design stage, it is necessary to evaluate the ultra-high cycle fatigue. NUREG provided the fatigue test results of nickel-based alloy materials used to manufacture heat transfer tubes [2]. The report used the test results to fit the fatigue average curve, and the fatigue design curve is obtained by dividing the stress amplitude by 2 and the number of cycles by 12. The conservatism of fatigue design curve considers the scatter of data, size effect, surface finish, and atmosphere. Although the ASME BPVC Section III criteria document [3] stated that these factors were intended to cover such effects as environment, Cooper [4] further stated that the term "atmosphere" was intended to reflect the effects of an industrial atmosphere in comparison with an air-conditioned laboratory, not the effects of a specific coolant environment. Therefore, numerous studies have been focused on the environmental degradations, such as stress corrosion cracking (SCC) [5–8] and corrosion fatigue [9,10]. These tests usually involved immersion tests of specimens with prefabricated defects in high-temperature water environment or low cycle fatigue tests in high-temperature water environment, without considering the influence of fretting load on fatigue performance.

Lykins [11] designed a device for fretting fatigue test and carried out fretting fatigue test of EA4T steel. Based on similar design, the references [12–14] carried out fretting fatigue tests on 690 alloy sheet specimens in normal temperature air or high temperature air and gave the influence of fretting on fatigue properties. Corigliano [15] applied the Static Thermographic Method during tensile tests and correlate the temperature trend to the fatigue properties of S335 steel. In reference [16], a fretting fatigue test under high temperature water environment was completed, and the coupling effect of fretting and corrosion was discussed. Since fatigue tests are usually carried out under low-cycle conditions ($10^3$–$10^5$), the effect of fretting on fatigue under high cycle conditions cannot be directly given.

The SWT model is widely used in fatigue failure analysis of multiaxial stress [17–19]. Ince [20] proposed an improved SWT model for different average stress levels. Fatemi et al. [21] believed that the shear strain model is more conducive to the fatigue analysis of multiaxial stress in low cycle fatigue. Luke [22] compared the use of SWT and FS models in fretting fatigue. Fouvry et al. [23] proposed the realization method of fretting wear in finite element calculation based on Archard model. On this basis, Tang [24] proposed a multi-layer grid adjustment method for the calculation of large depth fretting wear. Considering the simultaneous application of both wear calculation and fatigue calculation, this paper adopts the SWT model and the Archard model for simulation. In this paper, the effect of fretting on fatigue performance of heat transfer tube material 690 alloy is discussed by means of high cycle fretting fatigue test of tube specimen in room temperature air, low-cycle fretting fatigue test of sheet specimen in high-temperature water environment, and SWT fretting fatigue predicting simulation, and the conservatism of ultra-high cycle design fatigue curve is discussed.

## 2. High-Cycle Fretting Fatigue Tests on Tube Specimens

The mechanical property of a heat transfer tube is slightly different from that of the initial material after it is processed into a tube. Therefore, in the process of the fretting fatigue test, it is a priority to use the tube specimen of the heat transfer tube to carry out the test.

### 2.1. Test Devices and Parameters

In the tests, the 690-alloy tube of the nuclear steam generator heat transfer tube was used as the main research object, and the fretting fatigue test was carried out in the air environment.

The wall thickness of the 690-alloy tube is uniform, so when the axial load is applied, the stress level of the heat transfer tube is the same at all positions theoretically. In the mechanical properties tests, although adding a protective layer at the clamping position can solve the indentation problem, the stress concentration cannot be eliminated. Therefore, the fatigue crack will appear in the clamping part. In the tubular specimens of the fretting fatigue test, a clamping end protective design as shown in Figure 1a is taken to avoid the

connection between the device of hydraulic chuck and the pipe specimen so that stress concentration on the specimen can be limited as small as possible and the influence of the fretting on fatigue properties of materials can be reflected. Figure 1b shows the tube specimen after installation. The intermediate test section shall be protected with white stickers to ensure that the surface will not be damaged during installation.

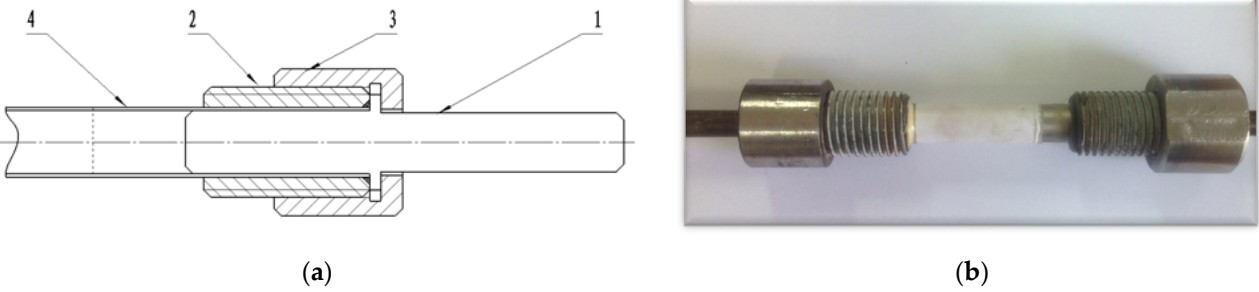

(**a**)                                                                (**b**)

**Figure 1.** Fretting fatigue test tube specimens: (**a**) specimen holding design (1. inner connecting staircase; 2. external thread casing; 3. inner threaded connection casing; 4. fretting fatigue test tube specimen); (**b**) picture of the specimen.

In this paper, a reliable test device [25] for lateral load of specimen is designed as shown in Figure 2a, so that the lateral load will not change when the specimen is subjected to axial load and relative sliding. The lateral loading device uses a fixed base with a guide groove, a horizontal slider structure, a cantilever structure with bearings, a fretting pair fixing device, and a lateral loading system to provide lateral load to the sample. The fixed base connects the device with the fatigue testing machine to provide rigid support. The horizontal slider structure and the cantilever beam structure with bearings ensure that the loads received on both sides of the tubular sample are equal. The lateral load loading system provides measurable contact loads. The lateral loading system is composed of a pressure sensor, a screw micrometer, a fixed plate and a screw connection device. During loading, the rotary screw micrometer provides quantitative displacement and compresses the pressure sensor to obtain the corresponding load. Due to the high accuracy (0.002 mm) of the screw micrometer, the applied lateral load also has an accuracy of 0.1 N.

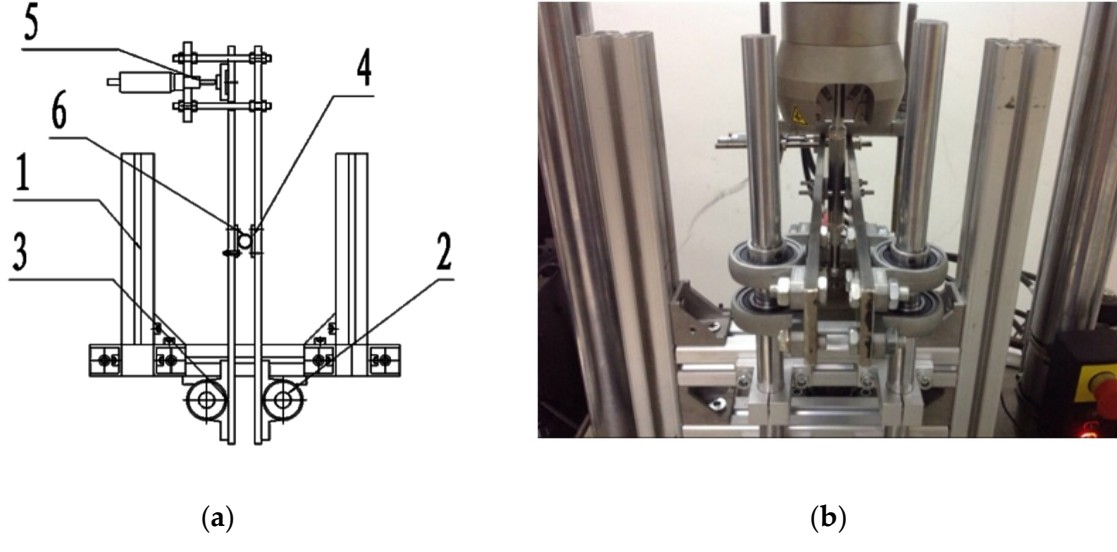

(**a**)                                                                (**b**)

**Figure 2.** Fretting fatigue test device for tube specimens: (**a**) specimen holding design (1. fixed base, 2. horizontal slider structure, 3. cantilever structure with bearings, 4. fretting pair, 5. lateral loading system, 6. tube specimen); (**b**) picture of the test device.

On this device, 690-alloy tube fretting fatigue tests under certain lateral force were carried out. The test was carried out mainly by load control, the stress ratio R = 0.1, the loading frequency was 20 Hz, the maximum axial load was 12 kN, 13 kN, 14 kN, 15 kN, 16 kN, and lateral load was 400 N for fretting fatigue tests to obtain the fatigue life of 690 alloy tube.

### 2.2. Test Results and Analysis

Figure 3 shows a sample fracture photo of the specimen in the fretting fatigue test. It is shown that the fretting fatigue crack often occurs at the end of the wear mark. Due to the cylinder–plane contact configuration between the heat transfer tube and the anti-vibration bar, the maximum contact pressure appears on both ends of the contact surface, which is also the most likely location for fretting fatigue cracks initiation. The fracture of the tube is studied by scanning electron microscope as shown in Figure 4. It is illustrated clearly that the crack initiates from the contact point with the AVB (anti-vibration bar) and grows to the other parts of the section.

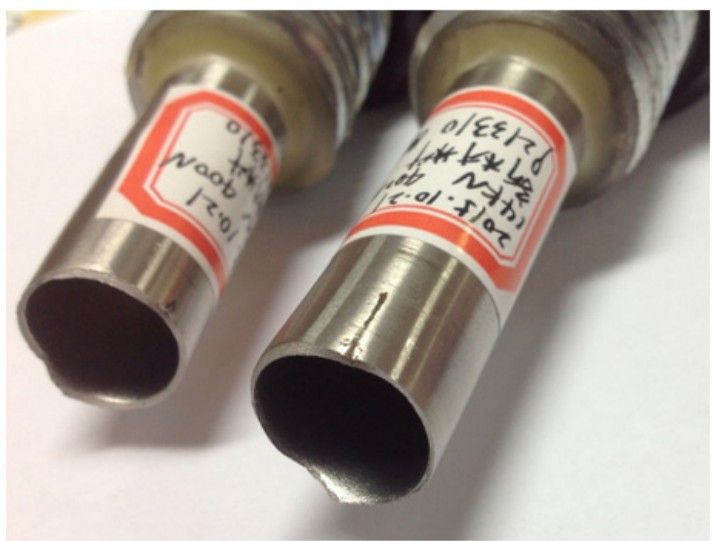

**Figure 3.** Fracture morphology of the fretting fatigue tube specimen.

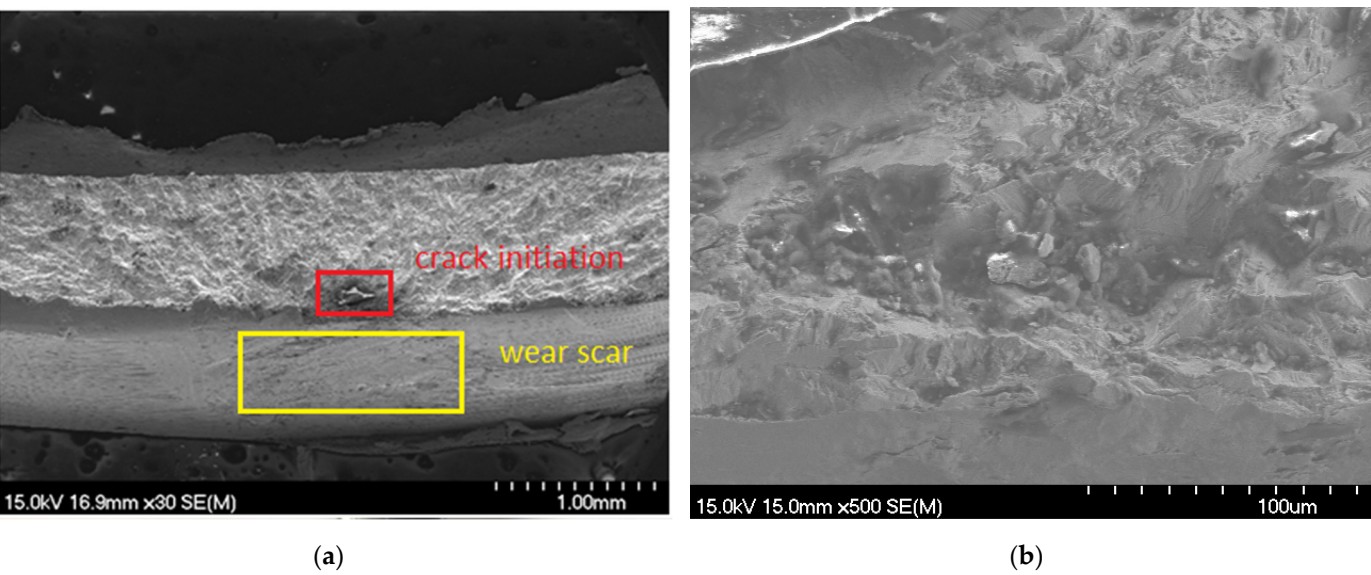

**(a)**                                                                                                       **(b)**

**Figure 4.** SEM of crack initiation and propagation of fretting fatigue:**(a)** section of tube; **(b)** crack initiation region.

Figure 5 shows the results of the comparison between the fretting fatigue test data and the ASME austenitic stainless steel design curve. It can be seen in the figure that there are three repeated results under each load. The discreteness does not exceed that of common fatigue tests, and the repeatability of the test is guaranteed. Because strain control (displacement control) is used in some tests, strain amplitude is used as Y axis in all S-N curves in this paper. The ASME curve provides the stress amplitude, where the corresponding strain amplitude is obtained by dividing by Young's modulus. The results show that, under the 400 N lateral force, the fatigue performance of the heat transfer tube is obviously decreased from the mean curve. However, the decreasing is less than the 800 N lateral force condition in the reference [14]. Under the condition of 400 N lateral load, the data points obtained by the fretting fatigue test are still above the design curve of ASME.

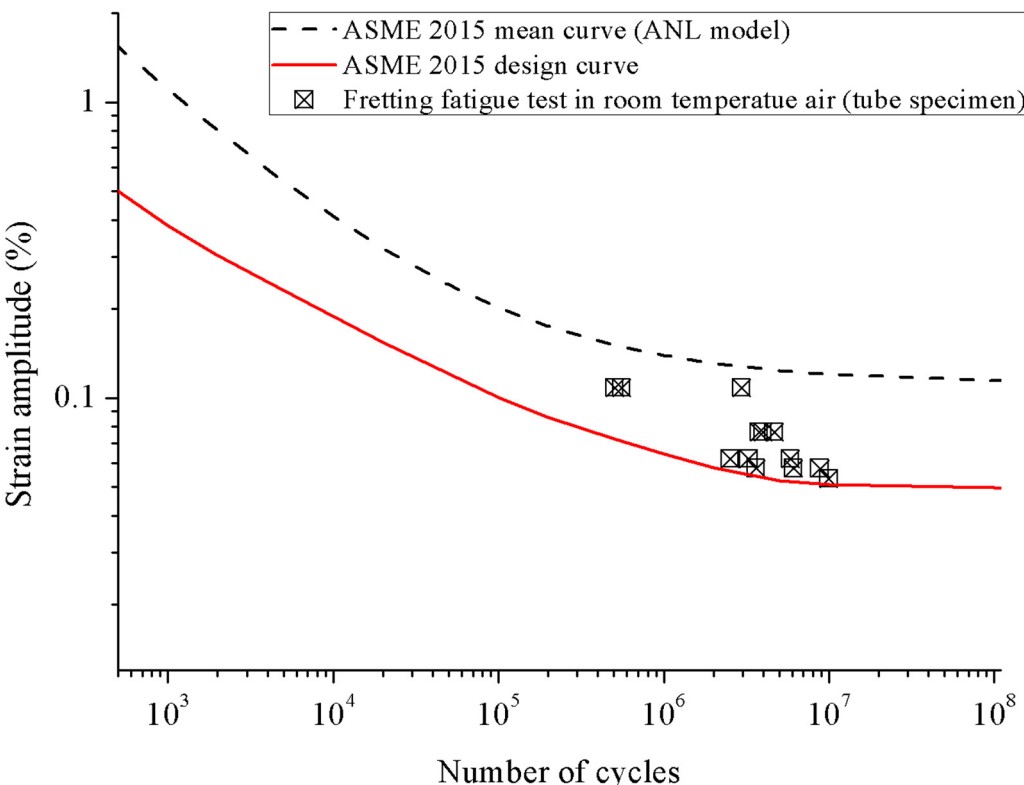

**Figure 5.** Fretting fatigue test data compared with the ASME design curve.

## 3. Low-Cycle Fretting Fatigue Tests on Plate Specimens in High-Temperature Water

It is generally believed that fretting only affects the fatigue performance of materials at high cycle. However, the influence of high-temperature water environment is the main factor that influences fatigue performance at low cycle. Fretting is a coupling effect between mechanical wear and electrochemical effect. Therefore, the fretting fatigue test under low cycle and high temperature water environment is designed, and the effect of fretting on fatigue performance under this environment is analyzed. At the same time, comparing with the fatigue design curve of ASME, the conservatism of the design curve is discussed. Considering the difficulty of clamping and lateral force application of tubular specimens under high temperature water, the test was carried out with 690-alloy sheet specimens. The sheet specimen is obtained from the 690-alloy tube blank, which is the upstream component of the heat transfer tube, which is close to the heat transfer tube in the microstructure.

### 3.1. Test Devices and Parameters

Based on the corrosion fatigue testing device for the high temperature and high-pressure circulating water of nuclear power, a positive pressure device is added to the high-temperature water environment fretting test device. The device consists of four parts,

which are high-temperature and high-pressure circulating water circuit, high-pressure kettle, SHIMADZU electro-hydraulic servo fatigue testing machine, and control system. The same water environment control system is used in the references [6,7,9,10], and will not be repeated in this paper. The heat caused by deformation and fretting friction during the fatigue test can be well removed by using this water circuit.

Figure 6 shows the installed device providing the lateral load in the fretting fatigue test [26]. Compress the spring by rotating the bolt at the end of the spring and maintain the compression to provide lateral load. The spring stiffness is measured under high temperature, and the compression is determined according to the spring stiffness to maintain the lateral load. After the test, re-measure the spring stiffness and calibrate the lateral load. The actual lateral load will be less than the expected applied load by no more than 10% due to high-temperature creep relaxation, vibration relaxation, etc.

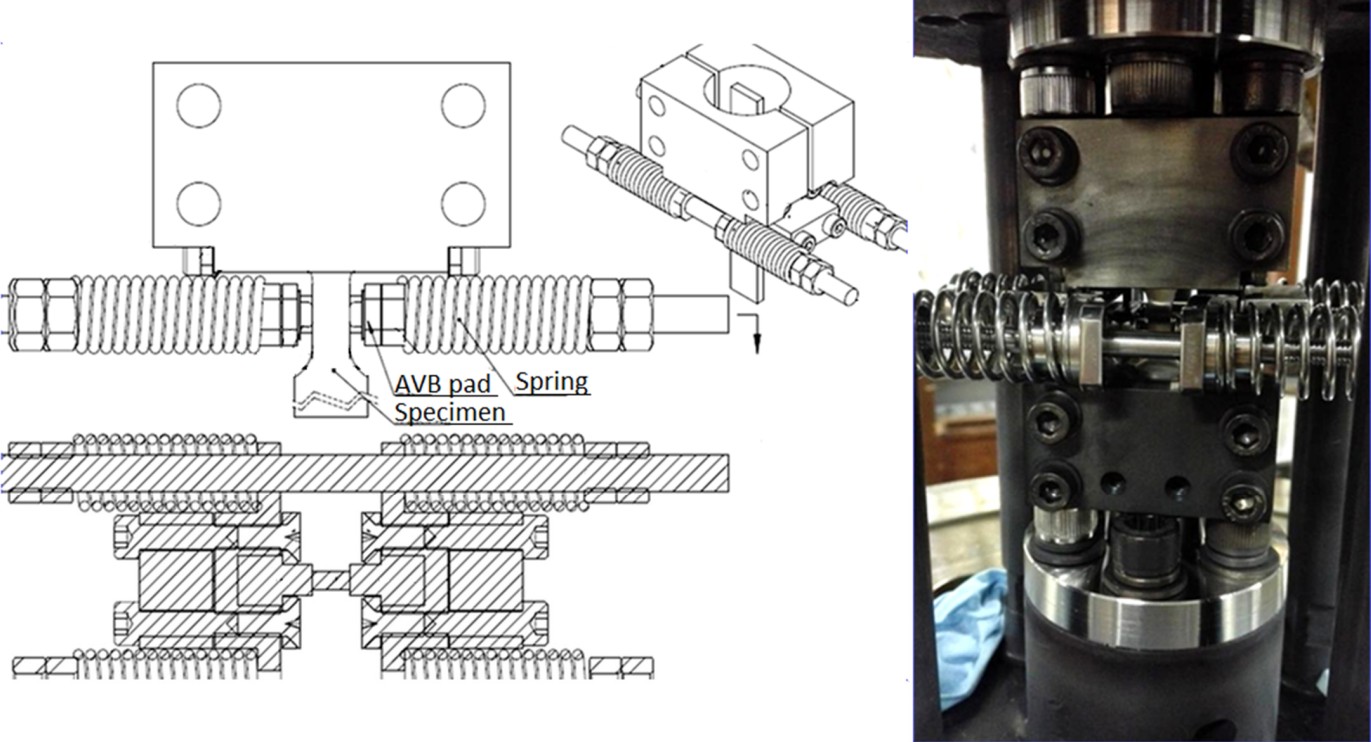

**Figure 6.** Normal pressure applied device for fretting fatigue test.

The S-N data points are obtained under the condition of 285 °C high-pressure water, a lateral force of 100 N, and a strain rate of 1% s$^{-1}$. The strain amplitudes are set from 0.1% to 0.45%, and the dissolved oxygen is controlled under 5 ppb.

At the same time, 0.25% and 0.4% amplitudes were selected for the comparative fretting fatigue test in normal temperature air and the lateral force was also 100 N and the strain rate was 1% s$^{-1}$.

### 3.2. Test Results and Analysis

The sheet sample used in the test was taken from a 690-alloy heat transfer tube blank with an outer diameter of 63.1 mm and a wall thickness of 10 mm. Tube blank is the intermediate product in the industrial production process of heat transfer tubes. The chemical composition of the tube blank was analyzed, and the tested area was located in the middle area of the 690-alloy tube blank. The chemical composition analysis results are shown in Table 1. The analysis results show that the element contents of 690-alloy heat transfer tubes are basically within the standard composition range. The material composition is consistent with that in Section 2.

**Table 1.** Chemical composition of the tube blank.

| Chemical Composition | Reference Value | Measured Value |
|:---:|:---:|:---:|
| C | 0.01–0.04 | 0.025–0.035 |
| N | - | 0.013 |
| Cr | 28–31 | 29.73 |
| Fe | - | 9.5 |
| Mn | ≤0.50 | 0.29 |
| P | ≤0.025 | 0.007 |
| Si | ≤0.50 | 0.29 |
| S | ≤0.015 | <0.001 |
| Al | - | 0.20 |
| Ti | - | 0.20 |
| Cu | - | <0.01 |
| Nb | - | <0.01 |
| Co | ≤0.10 | <0.01 |
| Ni | ≥58 | 58.7 |

　　Figure 7 shows the metallographic structure in three orientations of 690-alloy heat transfer tube blank for steam generator, which is a typical austenitic structure. The grain size is relatively uniform, and the average grain size is about 50 μm. It is also basically consistent with the materials in Section 2.

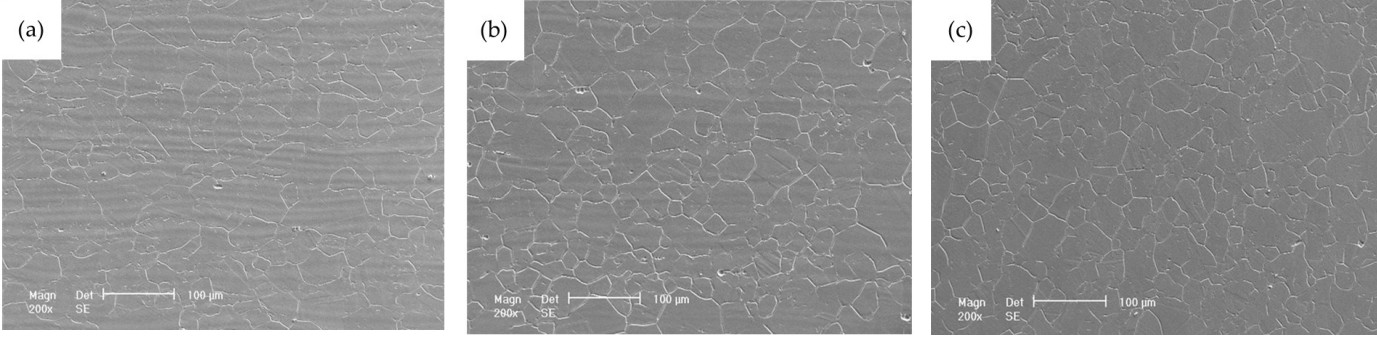

**Figure 7.** The metallographic structure in three orientations: (**a**) axial direction; (**b**) radial direction; (**c**) circumferential direction.

　　Figure 8 shows the photos of the specimen after fatigue failure. The fatigue crack starts at the edge of the fretting contact point and extends inward. Figure 9 shows the fracture morphology of the sample after knocking. The fracture surface is relatively straight, the crack starts on the surface of the fatigue sample and presents a typical fan-shaped pattern. There is a "ridge" on the fracture surface, which is a multi-crack initiation feature (Figure 9a); The fatigue crack growth zone has typical fatigue striation characteristics (Figure 9b,c), which is caused by repeated passivation and sharpening of the fatigue crack tip (Figure 9d). It is the knock off zone of fretting fatigue specimen in liquid nitrogen, showing typical dimple characteristics. Fine granular corrosion products were observed on the fracture surface of the fatigue specimen, indicating that the low cycle fatigue property of the material can be reduced by the mechanochemical interaction in the high-temperature and high-pressure water environment.

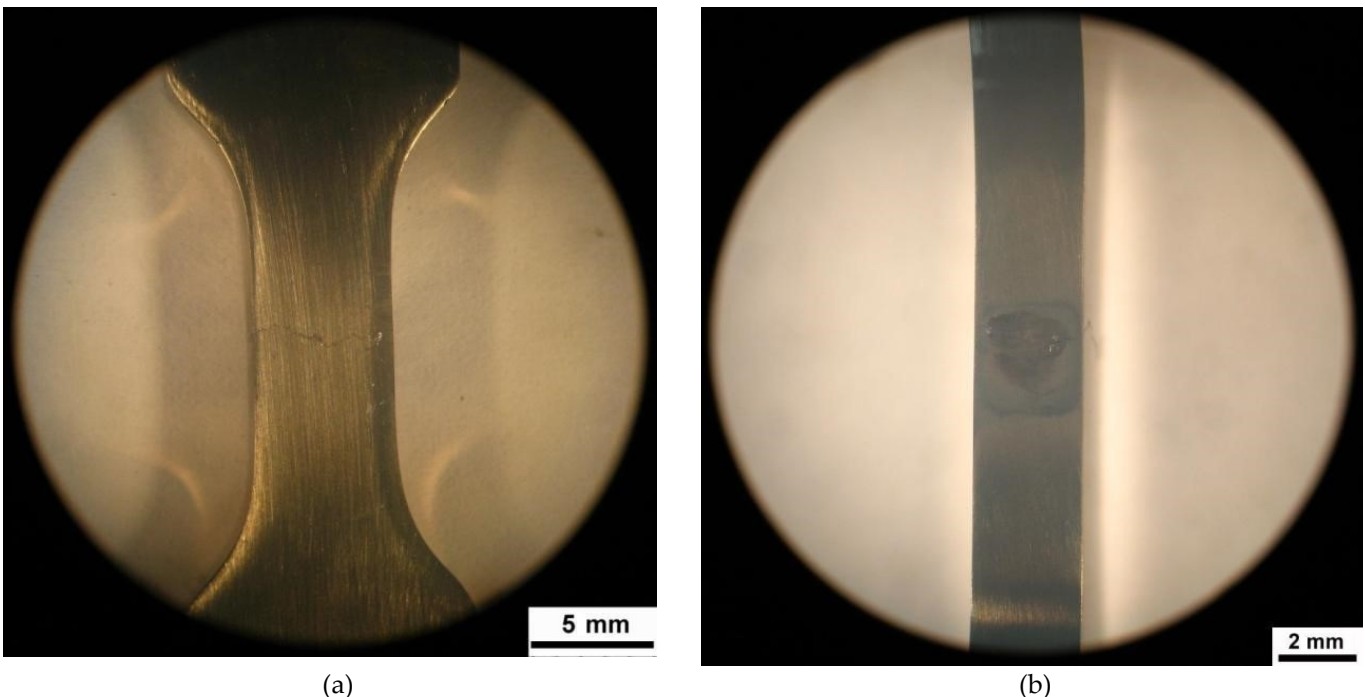

**Figure 8.** Photograph of specimen with fretting fatigue fracture; (**a**) front of the specimen; (**b**) side of the specimen (fretting contact position).

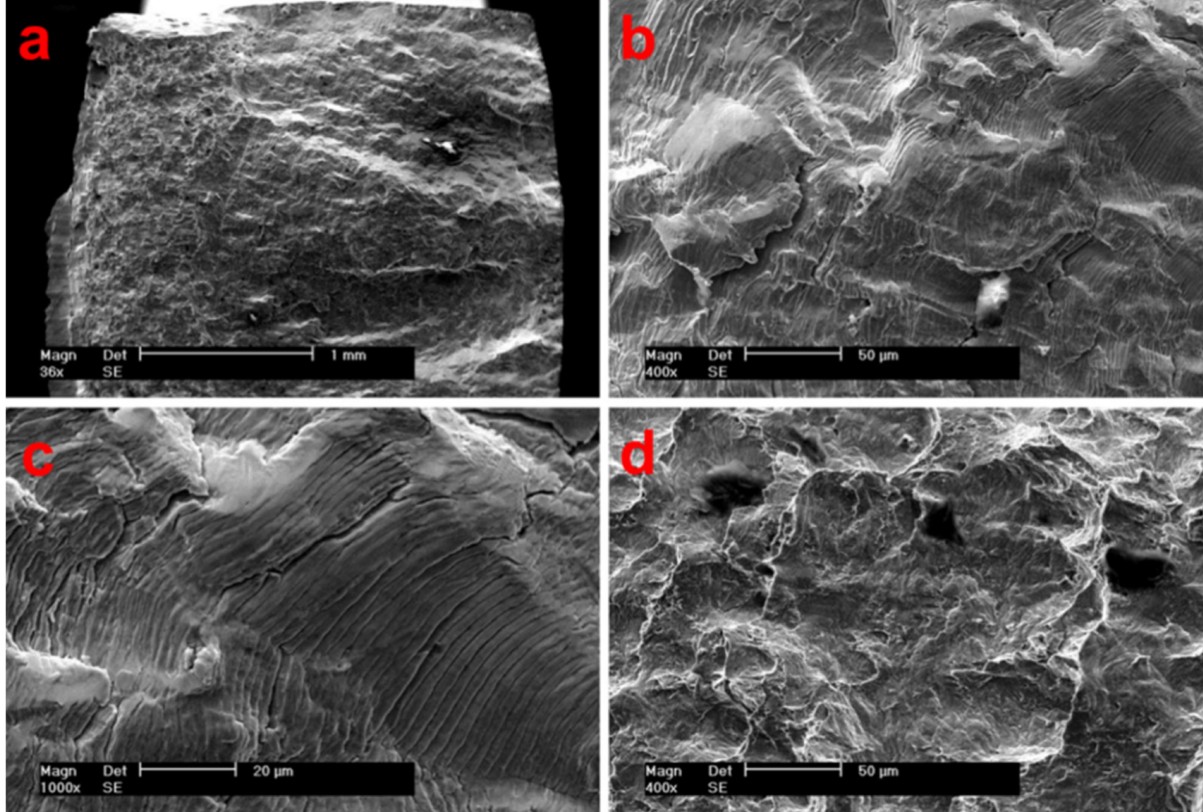

**Figure 9.** Fatigue fracture morphology: (**a**) Macro crack surface morphology; (**b**) crack propagation zone; (**c**) fatigue striation; (**d**) knock zone dimple.

The relationship between strain fatigue life and strain amplitude for 690-alloy at room temperature under 100 N lateral force is shown in Figure 7. It can be seen in the figure that

there are three repeated results under each load. The discreteness does not exceed that of common fatigue tests, and the repeatability of the test is guaranteed. The mean curve and design curve of austenitic alloy in ASME-BPVC III Appendix (version 2015) [27] are given in the figure. The figure shows that 0.4% strain amplitude fatigue data are located near the ANL mean curve, however the 0.25% strain amplitude of fatigue data is slightly lower than that of the mean curve. It is indicated that the lower the cycle times, the smaller the effect on the fatigue performance.

Figure 10 also gives the relationship between strain fatigue life and strain amplitude for 690-alloy in high-temperature and high-pressure water under 100 N lateral force. According to the graph, fretting fatigue data in high-temperature and high-pressure water are mainly distributed in the area below the ANL mean curve, which indicates that the fretting fatigue performance of the 690-alloy shows a definite environmental weakening effect. However, it can still be enveloped by the ASME design curve.

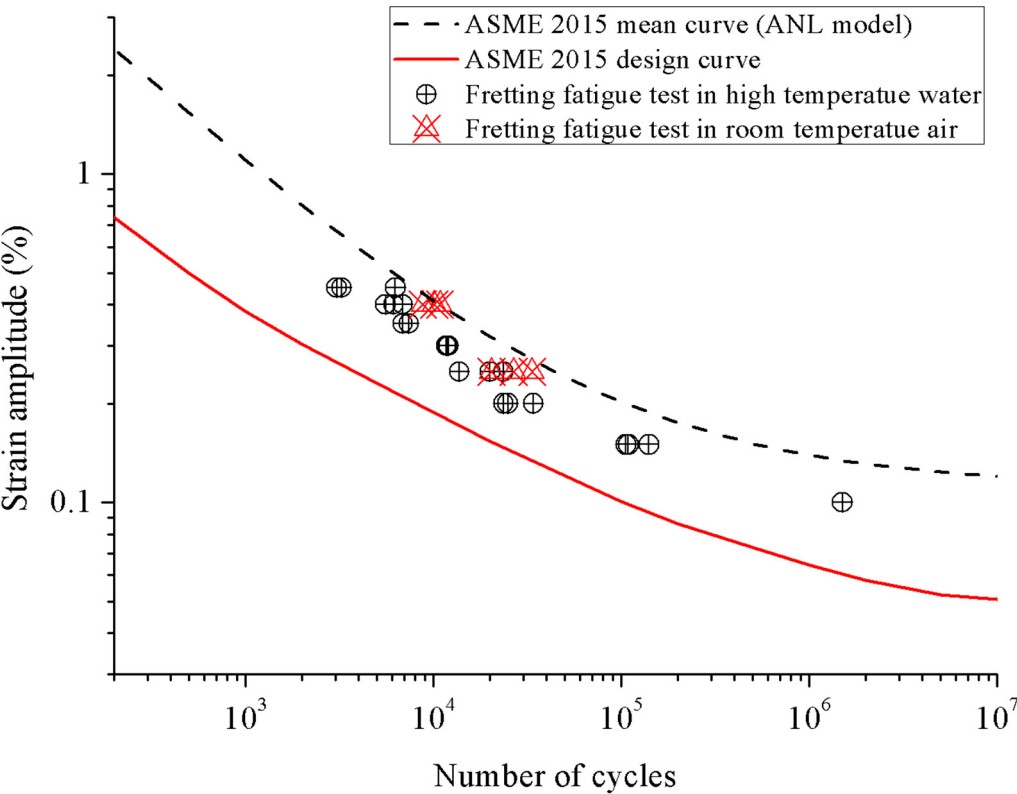

**Figure 10.** Fretting fatigue test results for 100 N pressure in normal temperature air and high-temperature water.

To distinguish between the effects of water environment on high temperature and fretting fatigue performance.

Figure 11 gives the results of fretting fatigue test in high-temperature and high-pressure water and the results of EAF (environmental assisted fatigue) reported by NUREG/CR-6909 [2,28], JNES-SS-1005 [29], and reference [10] The fatigue mean curve and design curve of Austenitic Alloy given by ASME-BPVC III Appendix (version 2015) are also included. It is shown that in high-temperature and high-pressure water, even in the 1%s$^{-1}$ high strain rate conditions, the fretting leads to a decrease in the fatigue life of 690 alloy, indicating the synergistic effect of fretting wear and high temperature and high-pressure water environment will lead to the 690 alloy showed the EAF effect. The fatigue performance of strain rate 1% s$^{-1}$ under fretting was close to the environmental impact fatigue test results at strain rate 0.1% s$^{-1}$ to 0.001% s$^{-1}$.

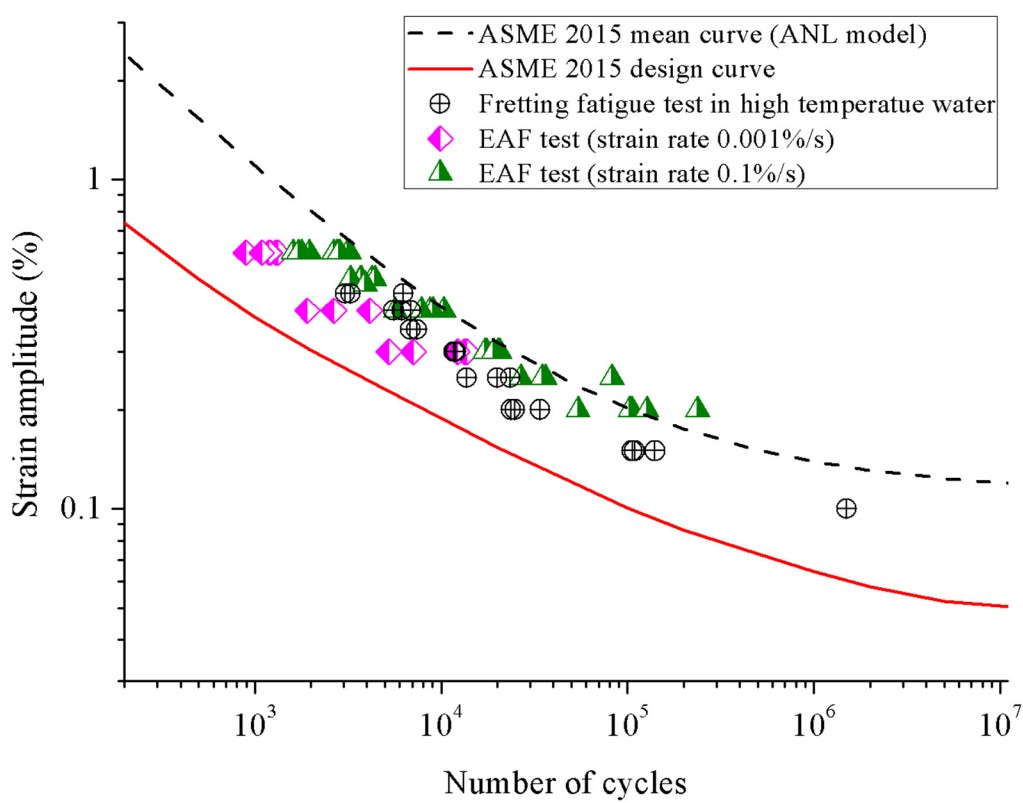

**Figure 11.** Fretting fatigue test results for 100 N pressure in high temperature water compared with the EAF (environmental assisted fatigue) test results reported by references.

## 4. Fretting Fatigue Simulation for Ultra High Cycle

It is proven that the fatigue design curve is conservative in the low and high cycle range through the fretting fatigue test. However, the flow induced vibration load on the heat transfer tube will reach $10^{10}$ to $10^{12}$ times in the life period. Therefore, it is very important to obtain the influence of fretting on the fatigue curve of the heat transfer tube material under the ultra-high cycle condition by numerical simulation.

### 4.1. SWT Method

The practical engineering components generally undergo a variety of cyclic loads, which are usually given in the form of a list of load amplitude-times. If the effect of the loading sequence is ignored, the cumulative fatigue damage can be predicted by the Palmgren–Miner rule:

$$D_k(x) = \sum_{i=1}^{k} \frac{N_i}{N_{fi}(x)} \tag{1}$$

where $N_i$ is the times of load $i$, $N_{fi}$ is the number of failures on the fatigue curve according to the stress amplitude under load $i$, $k$ is the length of the load list, and $x$ is the length along the structure. For the components under the condition of alternating multi-axis stress–strain, the strain energy dependent fatigue criterion can be considered as the effect of stress and strain. The SWT model is used in this paper, such as formula (2).

$$\text{SWT} = \left( \sigma_n^{max} \frac{\Delta \varepsilon_n}{2} \right)_{max} \tag{2}$$

where $\sigma_n^{max}$ is the maximal stress in direction n during the load cycle, $\Delta \varepsilon_n$ is the amplitude of strain in direction n. Take the maximum value of traversal product for all directions as SWT. The specific calculation process of the SWT parameters of one element is as follows: Firstly, the stress and strain of the element in a loading cycle under fretting are obtained by

the finite element calculation; secondly, the normal stress $\sigma_n(t)$ and normal strain $\varepsilon_n(t)$ in each direction $n$ in a cycle period are calculated; then, the maximal normal stress in one cycle $\sigma_n(max)$ and the strain amplitude $\Delta\varepsilon_n$ are calculated; and the maximal product of all directions is the SWT parameter, as shown in formula (2).

The corresponding failure times $N_f$ can be obtained by replacing the SWT parameters and the material fatigue parameters $\sigma_f$, $\varepsilon_f$, $b$, and $c$ into formula (3).

$$\left(\sigma_n^{max}\frac{\Delta\varepsilon_n}{2}\right)_{max} = \frac{\left(\sigma_f\right)^2}{E}\left(2N_f\right)^{2b} + \sigma_f\varepsilon_f\left(2N_f\right)^{b+c} \tag{3}$$

*4.2. FEM Model and Simulation Results*

ABAQUS software (V6.12, DASSAULT Systems, Waltham, MA, USA) was used for finite element calculation in this paper. A two-dimensional finite element model, as shown in Figure 12a, was set up, and Figure 12b shows the detailed mesh division of the finite element model at the contact. Two-dimensional plane-strain plate element was used in the calculation, Young's modulus was set as 191 GPa, Poisson's ratio was 0.3, the contact method was the penalty function method, and the tangential friction coefficient was 0.6. According to the Hertz contact calculation, the contact half width of the arc area was 35 μm, and the local contact area used 5 μm elements, which can ensure the mesh convergence.

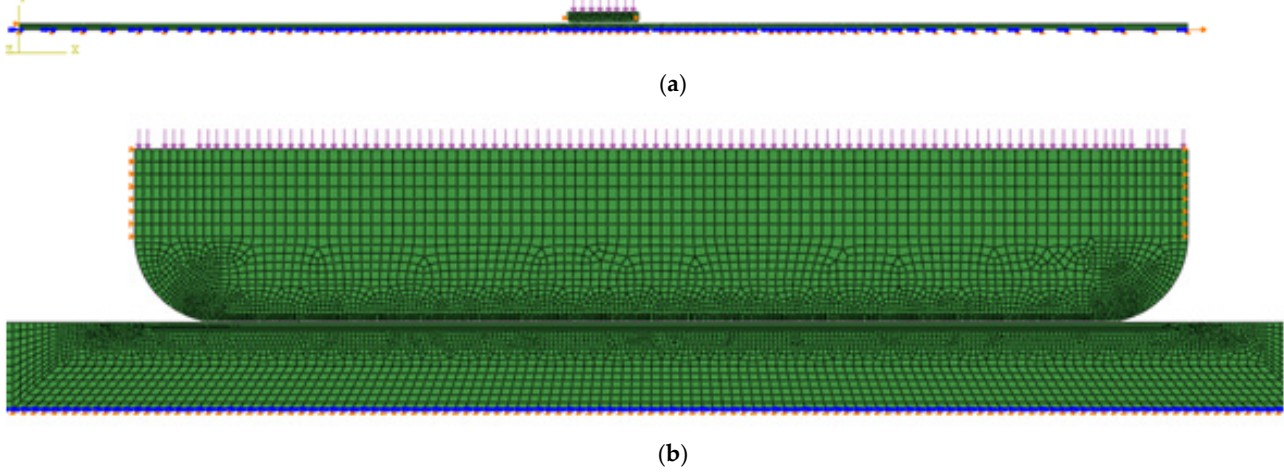

(a)

(b)

**Figure 12.** FEM model: (**a**) whole model; (**b**) detailed mesh of the contact region.

The X direction displacement was imposed on both sides of the pressure head. A symmetric boundary condition was applied to the left end and below the specimen, and the displacement boundary conditions were applied to the right end of the specimen according to the needs of different stress amplitudes in Figure 10. The friction coefficient on the contact surface was set to 0.6. The pressure load on the back of the pressure head was applied to simulate the clamping effect of the anti-vibration bar on the heat transfer tube. A multi-layer mesh update method introduced in reference [24] was used to modify the worn surface. In this paper, the fatigue curves under the fretting condition were calculated with 50 N and 100 N as contact pressure.

The calculation of each point on Figure 10 was carried out according to the following process:

(1) Establish the model in Figure 12, determine the stress amplitude ($\Delta\sigma$) and contact pressure, and calculate the $i^{th}$ cycle;

(2) Calculate SWT and $N_{fi}$ of each element according to the method in Section 4.1;

(3) Determine the wear amount under the number of $N_i$ cycles according to the method of [24], and update the surface geometric model;

(4) Calculate $D_k$;

(5) If $D_k < 1$, $i = i + 1$, return to (1); if $D_k$ of any element reaches 1, calculate $\sum N_i$;

(6) Draw point according to $\Delta\sigma$ and $\sum N_i$ in Figure 10.

As shown in Figure 13, the fretting fatigue S-N curve under the contact pressure of 50 N and 100 N of the heat transfer tube material was compared with the mean curve and the design curve of the material as well as test results. The simulation predicted a more conservative fatigue property than the tests in low cycles under same lateral forces because of no consideration of wear. This indicates that the fatigue curve of the material under the lateral force is obviously lower than the mean curve, and the higher the lateral force is, the lower the fatigue curve. When the lateral force rises to 100 N, the fatigue curve under the fretting condition appears below the design curve in the ultra-high circumferential region. This also shows that it is necessary to modify the fatigue curve in ultra-high cycle conditions.

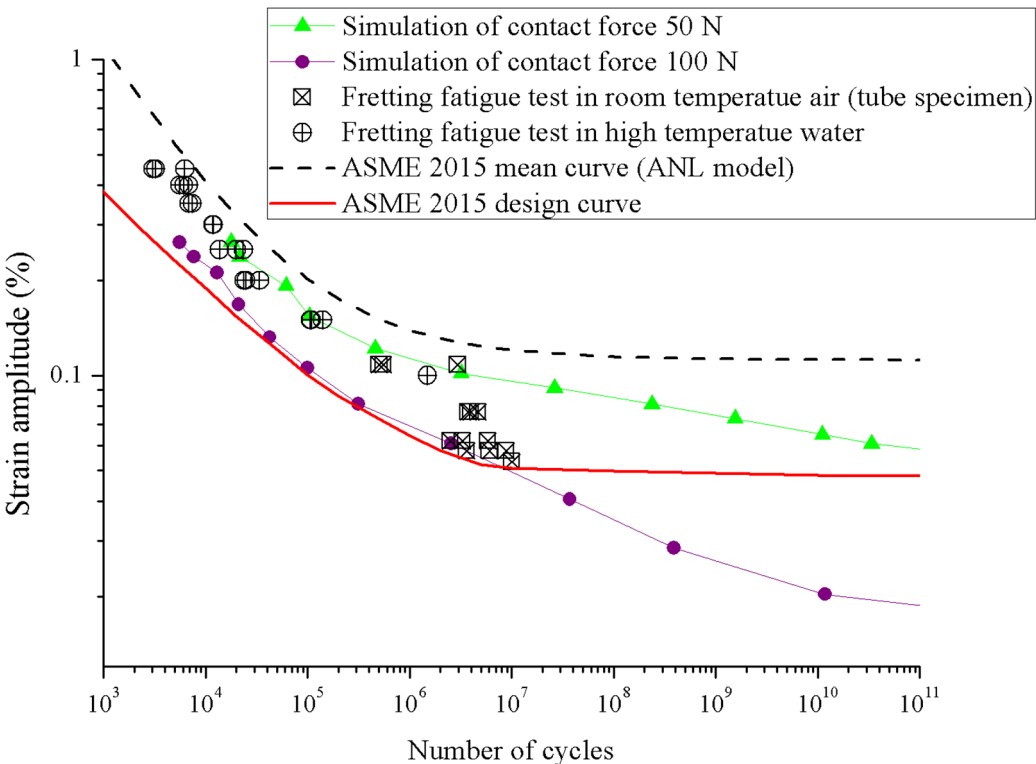

**Figure 13.** Fretting fatigue predicting curve compared with mean curve and design curve.

## 5. Conclusions

In this paper, the fretting fatigue test under normal temperature conditions is carried out for tubular specimens of 690 alloy of steam generator heat transfer tube. The test data were given in the range of high cycle number ($10^5$–$10^7$), and the results were compared with the mean curves and design curves of ASME. It is shown that, in this range, the fatigue strength is lower than the mean curve 20–50%, but it can still be enveloped by the design curve of ASME.

The fretting fatigue test of 690-alloy sheet specimens under high temperature water environment is introduced in this paper. The test data were given in the range of low cycle times ($10^3$–$10^5$), and the results were compared with ASME's mean curve and environmental assisted fatigue test results. Theoretically, the effect of fretting on the fatigue performance is very small under low cycle conditions. This shows that under the high-temperature water environment, the fretting also affects the fatigue performance under low cycle conditions. Even compared with the environmental assisted fatigue test at the same strain rate, the fatigue strength decreased significantly.

The flow induced vibration is a high frequency vibration, and it can reach $10^{10}$–$10^{12}$ times in the life period of the steam generator for dozens of years. Therefore, a SWT method is used to predict the S–N curve under ultra-high cycle conditions. The results show that under the condition of ultra-high cycle, the design curve of ASME can no longer envelop the effect of fretting on fatigue performance, so a further downward modification is necessary to ensure the safety of design.

## 6. Patents

Based on the work in Section 2, we applied for an invention patent in China: A symmetrical linear contact fretting fatigue test with micro-dynamic load loading device, Chinese patent, CN105510118A.

Based on the work in Section 3, we applied for an invention patent in China: A fretting fatigue test device with high-temperature and high-pressure circulating water and its application, Chinese patent, CN106990004A.

**Author Contributions:** Conceptualization and methodology, L.T.; project administration, H.Q.; simulation, C.L.; high-temperature experiment, X.W. All authors have read and agreed to the published version of the manuscript.

**Funding:** This research and the APC were both funded by Shanghai Rinsing-Star Project, grant number 20QB1403000.

**Data Availability Statement:** Not applicable.

**Acknowledgments:** Thanks to the team of Linjun Xie from Zhejiang University of Technology. They generously provided room temperature fatigue test equipment for this paper.

**Conflicts of Interest:** The authors declare no conflict of interest.

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
