# Peer review of "Fretting Fatigue Test and Simulation Analysis of Steam Generator Heat Transfer Tube"

_metals, doi:10.3390/met13010067_

Round 1
Reviewer 1 Report
The paper deals with the fretting fatigue test and simulation of a Steam Generator heat transfer tube
There are some important issues to be addressed before the paper can be recommended for publication.
The description of the device reported in figure 2 must be clarified. I suggest adding a new figure with the correct description of the four components.
It is unclear how the pressure is controlled to simulate the fretting phenomenon. The authors seem to have applied 400N but did not clarify how they performed this task.
In Figure 7, again, it needs to be clarified how the load and the strain rate are controlled during the tests. Please describe all the equipment used and improve the draft by highlighting the significant parts.
The paper's more critical issue lies in analysing the fretting cracks that should be observed before the specimens are broken to prove they are actually referred to as the fretting phenomenon. No investigation seems to have been made on this point.
Please add more references on fretting fatigue and reduce the self-citation rate. I suggest having a look at the following recently published paper Lubricants 2022, 10, 53. https://doi.org/10.3390/lubricants10040053, where the authors can find many important references on fretting fatigue.
Author Response
The paper deals with the fretting fatigue test and simulation of a Steam Generator heat transfer tube
There are some important issues to be addressed before the paper can be recommended for publication.
The description of the device reported in figure 2 must be clarified. I suggest adding a new figure with the correct description of the four components.
It is unclear how the pressure is controlled to simulate the fretting phenomenon. The authors seem to have applied 400N but did not clarify how they performed this task.
Pictures and descriptions of tubular specimen test apparatus are added. Line 104-117
In Figure 7, again, it needs to be clarified how the load and the strain rate are controlled during the tests. Please describe all the equipment used and improve the draft by highlighting the significant parts.
High temperature water environment test device and load control mode are added. Line 184-189
The paper's more critical issue lies in analysing the fretting cracks that should be observed before the specimens are broken to prove they are actually referred to as the fretting phenomenon. No investigation seems to have been made on this point.
The sample analysis after high temperature water environment experiment is added to prove that the fatigue crack is really caused by fretting. Line 199-237
Please add more references on fretting fatigue and reduce the self-citation rate. I suggest having a look at the following recently published paper Lubricants 2022, 10, 53. https://doi.org/10.3390/lubricants10040053, where the authors can find many important references on fretting fatigue.
In Chapter 1, the relevant references on fretting fatigue is added. Line 61-77
Reviewer 2 Report
Considering that low, high and ultra hig cycle fatigue are mentioned and some of them are analyzed, a deeper discussion of the state of the artis suggested. Different methods are then used for the Fatigue life predictioons:
Only as examples, but many other are present:
Local Strain Estimation Method for Low- and High-Cycle Fatigue Strength Evaluation. Int. J.
Fatigue 2012, 40, 1–6.
A Structural Strain Method for Low-Cycle Fatigue Evaluation of Welded Components. Int. J. Press. Vessel. Pip. 2014, 119, 39–51
Recent Developments and Future Challenges in Fatigue Strength Assessment ofWelded Joints. Proc. Inst. Mech. Eng. Part C: J. Mech. Eng. Sci. 2015, 229, 1224–1239.
Considering the high thermal increment in this kind of tests, a consideration about monitoring the tests by fotrthe temperature detection would be appreciated. As an example, the following paper is suggested.
https://www.sciencedirect.com/science/article/pii/S2452321619303750?pes=vor
A figure of the tested specimen would be appreciated.
Figure 1b is of bad quality. It needs to be improved.
Figure 4 could be woring. The y axes does not report the stress amplitude is divided by S0 for normalization. The same for figs 7-8 etc.
Figure 5 is of bad quality. It needs to be improved. A deeper description of figure 5 is needed
The symbols should be introduced before using them. i.e symbols in eq 2 should be introduced before or right after the equation. They are described far from eq 2.
A description of the FE model should be more accurate. i.e. type of software used, element types, mechanical properties used, element dimension, convergence analysis (was it carried out?)
Figure 9b should be enlarged.
Author Response
Considering the high thermal increment in this kind of tests, a consideration about monitoring the tests by fotrthe temperature detection would be appreciated. As an example, the following paper is suggested.
https://www.sciencedirect.com/science/article/pii/S2452321619303750?pes=vor
In Chapter 1, the relevant references on fretting fatigue is added. Line 61-77
A figure of the tested specimen would be appreciated.
Added Figure 1 and related description. Line 99-101
Figure 1b is of bad quality. It needs to be improved.
Added Figure 2 and related description. Line 104-115
Figure 4 could be woring. The y axes does not report the stress amplitude is divided by S0 for normalization. The same for figs 7-8 etc.
All normalization processing is canceled and strain amplitude is used for drawing. Figure 5/10/11/13 and line 145-151
Figure 5 is of bad quality. It needs to be improved. A deeper description of figure 5 is needed
Figure 5 is a common water quality control method for loop of nuclear power plant, which is also used and described in other papers. This is not the focus of this article. Figure 5 has been deleted and references to relevant reference documents have been added.
The symbols should be introduced before using them. i.e symbols in eq 2 should be introduced before or right after the equation. They are described far from eq 2.
Eq2 related parameter description has been added. Line 287-289
A description of the FE model should be more accurate. i.e. type of software used, element types, mechanical properties used, element dimension, convergence analysis (was it carried out?)
Added detailed model description. Line 299-306
Figure 9b should be enlarged.
Figure 9b has been enlarged. Figure 12
Reviewer 3 Report
The authors of the study presented investigation of fretting on fatigue performance of heat transfer tube material 690 alloy, by means of high cycle fretting fatigue test of tube specimen in room temperature air, low cycle fretting fatigue test of sheet specimen in high temperature water environment and Smith-Watson-Topper fretting fatigue predicting simulation. The research seems to be reasonably well performed, but some points need to be addressed.
Based on the information contained in the study, samples in the form of a tube and a sheet of metal were selected for testing. At the same time, it was noted that these materials did not come from the same place that is subject to fretting fatigue. Please explain whether microstructure tests of selected elements were carried out before the test, and if so, the obtained results should be included in the study, especially since the microstructure of the material can significantly affect its fatigue resistance.
Please explain how the constant value of the lateral forces has been ensured, especially in the case of high-temperature tests. If the value of the force was determined at the beginning of the test, how was the influence of temperature change on the obtained results taken into account?
Please explain for each of the experimental methods used, how the repeatability of the instrumentation asembly with respect to the sample was ensured. Moreover, please provide information on how many samples experimental tests were carried out (for each load level).
Author Response
Based on the information contained in the study, samples in the form of a tube and a sheet of metal were selected for testing. At the same time, it was noted that these materials did not come from the same place that is subject to fretting fatigue. Please explain whether microstructure tests of selected elements were carried out before the test, and if so, the obtained results should be included in the study, especially since the microstructure of the material can significantly affect its fatigue resistance.
SEM analysis and discussion of sheet samples are added.
Please explain how the constant value of the lateral forces has been ensured, especially in the case of high-temperature tests. If the value of the force was determined at the beginning of the test, how was the influence of temperature change on the obtained results taken into account?
The description of two methods of applying lateral load for test is added. Line 217-237
Please explain for each of the experimental methods used, how the repeatability of the instrumentation asembly with respect to the sample was ensured. Moreover, please provide information on how many samples experimental tests were carried out (for each load level).
The description of test repeatability is added. Line 143-145 239-241
Round 2
Reviewer 1 Report
The paper is now acceptable for the publication in Metals.
Author Response
Thank you.
Reviewer 2 Report
The authors have improved the paper. However some issues still need to be solved. i.e. A deeper discussion on relevant literature, including thermal increment during fretting fatigue. The possibility of monitoring the tests using dedicated instrumentations could be also discussed (such as thermal cameras). In addition the english should be improved.
Author Response
Dear reviewer,
We have read the reference about Thermographic analysis during tensile tests and fatigue assessment. It was an interesting finding by using Thermographic method during static tests to predict its fatigue life.
Actrually, we also worried about the heat caused by deformation and fretting friction during fatigue test. In the high temperature water test, the heat will taken by the flow. In the room temperature air test, the temperature on the surface of specimen will reach 70~90℃。Considering that most of the room temperature fatigue tests will be completed under such conditions, we have not continued to discuss this issue.
the reference you recommonded is added in the paper, and also the discussion of the heat generation.
See Line 64-66 and 176-177.
And sorry, I am not a native speaker in English. :-)